# The Effectiveness of the Combination of Arterial Infusion Chemotherapy and Radiotherapy for Biliary Tract Cancer: A Prospective Pilot Study

**DOI:** 10.3390/cancers15092616

**Published:** 2023-05-05

**Authors:** Takuma Goto, Hiroki Sato, Shugo Fujibayashi, Tetsuhiro Okada, Akihiro Hayashi, Hidemasa Kawabata, Sayaka Yuzawa, Syunta Ishitoya, Masaaki Yamashina, Mikihiro Fujiya

**Affiliations:** 1Division of Metabolism and Biosystemic Science, Gastroenterology, and Hematology/Oncology, Department of Medicine, Asahikawa Medical University, 2-1 Midorigaoka-Higashi, Asahikawa 078-8510, Japan; hirokisato@asahikawa-med.ac.jp (H.S.); fujishu@asahikawa-med.ac.jp (S.F.); te1984@asahikawa-med.ac.jp (T.O.); hayashiakihiro0629@yahoo.co.jp (A.H.); kawa0527@asahikawa-med.ac.jp (H.K.); fjym@asahikawa-med.ac.jp (M.F.); 2Department of Diagnostic Pathology, Asahikawa Medical University Hospital, 2-1 Midorigaoka-Higashi, Asahikawa 078-8510, Japan; ysayaka528@asahikawa-med.ac.jp; 3Department of Radiology, Asahikawa Medical University, 2-1 Midorigaoka-Higashi, Asahikawa 078-8510, Japan; ishitoya@asahikawa-med.ac.jp (S.I.); ymasaaki@asahikawa-med.ac.jp (M.Y.)

**Keywords:** biliary tract cancer, gallbladder cancer, radiation therapy, arterial infusion chemotherapy, conversion therapy

## Abstract

**Simple Summary:**

Unresectable biliary tract cancer has a poor prognosis, with insufficient response rates from the standard treatment. We developed a new combination therapy regimen of intra-arterial chemotherapy plus radiation therapy based on a retrospective study, and we found its high effectiveness in cases of some unresectable biliary tract cancers. Then, we conducted this prospective study, which demonstrated the benefits for some patients with unresectable biliary tract cancer, resulting in high clinical response and disease control rates of 71.4% and 100%, respectively. Furthermore, two patients converted to surgery, indicating its high potential as a preoperative therapeutic strategy to achieve a long-term prognosis. This treatment was considered tolerated due to the absence of treatment-related deaths. This prospective pilot study is an important step toward determining the usefulness and safety of intra-arterial chemotherapy plus radiation therapy, as well as its potential future utility.

**Abstract:**

The standard treatment of unresectable biliary tract cancer (BTC) has shown an insufficient response rate (RR). Our retrospective setting revealed that a combination therapy consisting of intra-arterial chemotherapy plus radiation therapy (IAC + RT) provided a high RR and long-term survival benefits in unresectable BTC. This prospective study aimed to test the effectiveness and safety of IAC + RT as the first-line therapy. The regimen included one-shot IAC with cisplatin, 3–6 months of reservoir IAC (5-FU and cisplatin, q/week), and 50.4 Gy of external radiation. The primary endpoints include the RR, disease control rate, and adverse event rate. This study included seven patients with unresectable BTC without distant metastasis, with five cases classified as stage 4. RT was completed in all cases, and the median number of reservoir IAC sessions was 16. The RR was 57.1% for imaging and 71.4% for clinical assessment, and the disease control rate was 100%, indicating a high antitumor efficacy, which allowed two cases to be transferred to surgery. Five cases of leukopenia and neutropenia; four cases of thrombocytopenia; and two cases of hemoglobin depletion, pancreatic enzyme elevation, and cholangitis were observed, but with no treatment-related deaths. This study revealed a very high antitumor effect with IAC + RT for some unresectable BTC, and it could be useful for conversion therapy.

## 1. Introduction

The incidence of biliary tract cancer (BTC) seems to be increasing globally. The global incidence of BTC has increased worldwide in recent decades (0.3–6 per 100,000 people per year) [1]. The mortality rate for BTC is also increasing globally, according to 2020 data, ranging from 1 to 6 deaths per 100,000 population per year, excluding the Asian region with the highest incidence (>6 deaths per 100,000 per year) [2]. The morbidity and mortality of BTC are relatively high in Japan [3], ranking sixth in cancer-related deaths, accounting for approximately 18,000 deaths per year [4]. BTC is often diagnosed in an advanced stage and has a poor prognosis (5-year survival rate of <30%) because of few effective treatments [5]; thus, an effective treatment strategy is urgently needed.

Surgical resection currently represents the only curative treatment for BTC, but 70% of patients are deemed unresectable at the diagnosis [6], and approximately half of the patients who underwent resections relapse within 1 year [7]; thus, the 5-year survival rate remains low [8]. Most cases are unresectable; thus, they are instead treated with systemic chemotherapy, such as gemcitabine and cisplatin (GC), but the response rate (RR) is inadequate at 20–30%, and the median overall survival (mOS) and progression-free survival (mPFS) are only approximately 1 year and 6 months, respectively [9,10,11,12]. High RRs have been reported with intra-arterial chemotherapy (IAC) [13,14,15] and radiation therapy (RT) [16,17,18], but the efficacy of IAC with RT for unresectable BTC has not been established. Recently, our retrospective study noted the high efficacy of the combination therapy regimen of IAC and RT (IAC + RT) (mOS: 15.4 months, mPFS: 14.3 months, and RR: 40.4%) in cases of unresectable BTC [19].

Therefore, the present study prospectively examined the usefulness and safety of IAC + RT as the first-line treatment for unresectable BTC.

## 2. Materials and Methods

### 2.1. Study Design

This study was conducted with the cooperation of three hospitals in Japan, and data were collected and analyzed at our hospital. This trial was initially designed as a prospective study involving 20 patients, conducted from October 2015 to March 2020, but was terminated in March 2019 due to a change in the law concerning clinical research in Japan.

This trial was approved by our hospital’s ethics committee. All patients were required to give their written informed consent before the therapy could be administered, and the trial was conducted following the Declaration of Helsinki.

### 2.2. Patients

Patients were eligible for the study if they were ≥20 years old and had received a histopathological or cytologic diagnosis of nonresectable BTC (intrahepatic or extrahepatic cholangiocarcinoma, gallbladder cancer, or ampullary carcinoma); had an Eastern Cooperative Oncology Group performance status of 0 or 1; and had an estimated life expectancy of >3 months. Other eligibility criteria were an adequate hematologic and biochemical function, including a white blood cell of ≥2500/mm^3^, neutrophil of ≥1500/mm^3^, platelet count of ≥100,000/mm^3^, total bilirubin level of ≤3.0 g/dL, alanine aminotransferase and aspartate aminotransferase of ≤150 IU/L, and serum creatinine level of <1.2 mg/dL. Additionally, the following items were used as exclusion criteria: resectable BTC, distant metastases other than liver or lymph node metastases, prior chemotherapy treatment for other malignancies, active gastrointestinal tract ulceration, difficulty in controlling bowel movements, pulmonary fibrosis or interstitial pneumonia, active infections, severe major organ complications (e.g., heart failure, renal failure, liver failure, bowel obstruction), poorly controlled diabetes mellitus, active overlapping cancers (synchronous overlapping cancers and iatrogenic overlapping cancers with a disease-free interval of <3 years), receiving continuous systemic steroid administration, serious mental disorders, and severe drug allergies that make the drug administration difficult. These criteria were confirmed within 14 days before registration.

Table 1 shows valuable pretherapy information.

### 2.3. Treatment

Figure 1 shows the IAC + RT therapy schedule. The protocol was based on our previous report [19], but the off-label drugs epirubicin and etoposide were excluded, and the irradiation method was adapted to the current mainstream. First, blood flow modification and one-shot IAC of CDDP at 50 mg were performed for angiography. One-shot IAC comprised CDDP and 20 mL of 0.5% saline and was administered as a 30 min infusion. External beam radiation was started approximately 1 week after one-shot IAC. External radiation used linac X-ray of ≥6 MV with multi-portal irradiation administered to the target as split doses of 1.8 Gy, 5 times a week, with a maximum dose of 50.4 Gy.

Biliary tract tumors and metastatic lesions with a minor axis of ≥10 mm were defined as gross tumor volume (GTV), and a margin of 5 mm was added to the GTV as the clinical target volume (CTV). An error due to respiratory fluctuation and patient fixation reproducibility was assumed, and the planning target volume was added to the CTV with a margin of roughly 10–15 mm. The reservoir system of IAC was embedded in the subcutaneous groin area simultaneously with external beam radiation initiation. A catheter was placed with the gastroduodenal artery (GDA) coil method, and the side hole was positioned near the common hepatic artery. Coil embolization of the right gastric artery was performed when its branches inflowed the hepatic artery. Metal coil embolization was properly performed if an anatomical variation was present in the hepatic artery bifurcation. IAC from the reservoir (r-IAC) was performed with FP therapy (5FU at 750 mg/m^2^ and CDDP at 10 mg/m^2^) once a week. r-IAC was administered as a total of 30 mL of CDDP with 0.9% saline over a 30 min infusion, followed by a total of 120 mL of 5-FU with 0.9% saline over a 120 min infusion. r-IAC was terminated in the case of uncontrolled adverse events, progressive disease (PD), or the patient’s or clinician’s choice. Patients with no disease progression at three months could continue with another three months of the r-IAC regimen.

This IAC + RT protocol was defined as ending within 6 months from its initiation. Dose changes and delays were tolerated for hematological toxicity, febrile neutropenia, renal dysfunction, peripheral neuropathy, hearing impairment, or causal non-hematological toxicity of grade ≥3. The standard chemotherapy was subsequently performed unless curative surgery was deemed possible after the IAC + RT.

One-shot IAC was administered via the common hepatic artery with cisplatin at 50 mg (Arrow A). External beam radiation therapy (ERT) was started approximately 1 week after 1-shot IAC. ERT used a linac X-ray of ≥6 MV, with multi-portal irradiation administered to the target as split doses of 1.8 Gy, 5 times a week, with a maximum of 50.4 Gy. The reservoir system of IAC was embedded in the subcutaneous groin area almost simultaneously with ERT initiation. A catheter was placed by the gastroduodenal artery (GDA) coil method (Arrow B), and the side hole was positioned near the common hepatic artery. IAC from the reservoir (reservoir IAC) was performed with FP therapy (5FU at 750 mg/m^2^ and CDDP at 10 mg/m^2^) once a week.

### 2.4. Assessments

Patients were examined at the start of every cycle to monitor their symptoms and adverse effects and assess their blood count and hepato-renal function. All patients underwent pre-treatment computed tomography (CT) within 1 month before enrollment. The antitumor efficacy was assessed by CT at 3 and 6 months in patients receiving this treatment following the Response Evaluation Criteria for Solid Tumors (RECIST) 1.1. The progression-free survival (PFS) endpoint was defined as the day when the PD by RECIST 1.1 was determined by tumor growth or the appearance of new lesions. The protocol completion rate was defined as the rate at which IAC + RT therapy was completed for ≥3 months, regardless of drug suspension or dose reduction. Patients were regularly seen at the hospital and continued to be evaluated for symptoms; adverse events; and progression of lesions by physical examinations, blood sampling, and CT after completing the study treatment.

### 2.5. Statistical Analyses

The primary outcome was the tumor response and adverse event rate, and the secondary outcomes were the PFS, OS, protocol treatment completion rate, and one-year survival rate. The trial was designed to have 80% power to detect an increase in the RR from 19% in patients receiving gemcitabine plus cisplatin [10] to 55% in patients receiving IAC + RT. We first conducted a superiority test in RR with a single-arm Phase 2 trial before proceeding with the randomized Phase 3 trial, considering that BTC is a carcinoma with a low occurrence frequency. We set the target number of cases to 20, and the detection power was 87%, even if 16 cases had a dropout rate of 20%, assuming from the above RR based on a single-proportion Fisher exact test with a one-sided significance level of 5%. Toxic effects were categorized following the National Cancer Institute’s Common Toxicity Criteria for Adverse Events, version 4. OS and PFS were analyzed with the Kaplan–Meier curves and the log-rank test. Patients who did not have PD and those who died were excluded at the date of their last follow-up. The database was closed for analysis in June 2019.

## 3. Results

We recruited 7 patients without initial treatment at our hospital from September 2015 to March 2019. Stage 4 accounted for the majority, being noted in five cases, and the other baseline characteristics are shown in Table 1. An anomaly in which the right hepatic artery (RHA) diverged from the superior mesenteric artery (SMA) was observed in one case. Hepatic arteriography confirmed a replaced RHA, and coil embolization of the RHA diverging from the SMA was performed to unify blood flow. A histological diagnosis of adenocarcinoma was obtained in all cases, with partial squamous differentiation observed in one patient. The median follow-up time was 9.4 months. No cases demonstrated tumor progression during IAC + RT, while three deaths occurred after IAC + RT completion.

### 3.1. Tumor Response

Objective tumor responses were measured in all cases and are shown in Table 2 and Table 3. An antitumor effect was achieved in all patients, with a partial response (PR) in four cases and stable disease (SD) in three cases (RR: 57.1%, disease control rate: 100%). Notably, one patient who was considered SD on preoperative RECIST was clinically judged with complete response (CR), as no residual cancer cells were detected on postoperative pathology, and the patient subsequently survived without recurrence until the end of the study. The tumor shrinkage rates with RECIST ranged from 14.5% to 53.6%, with a median of 32.4%. Clinically, the postoperative pathological examination of a patient scheduled for surgery after IAC + RT revealed no residual cancer cells, with a 100% shrinkage rate. Therefore, clinical tumor shrinkage rates ranged from 14.5% to 100%, with a median of 37.6%.

### 3.2. Treatment Compliance

Radiation therapy was completed in all cases. There was 1 patient who was unable to receive r-IAC beyond 3 months because the study was ended, but the regimen was completed by 6 patients. Therefore, the protocol completion rate was 85.7%. The median treatment duration was 5.5 months, and the number of intra-arterial injections ranged from 5 to 20, with a median of 16. The r-IAC was continued in all patients up to the end of the study, but regular weekly r-IAC was difficult, and treatment was postponed in all patients. Treatment was postponed in each case one to three times, with a median of two. The total number of postponements was 15 in all cases, with thrombocytopenia in 9 cases as the most common reason for the postponement, followed by leukopenia/neutropenia and cholangitis in three cases each. The transition to second-line treatment was attempted in four patients, including two with surgery and two with GC therapy. The remaining three patients had difficulty transitioning to second-line treatment, but two of them had just completed IAC + RT at the end of the study period; thus, only one patient had difficulty transitioning to second-line treatment.

### 3.3. Survival and Disease Progression

Table 3 shows the prognoses in all cases. The final analysis was performed 3 months after the last patient was enrolled in the study due to the influence of Japanese law, by which point 3 deaths had occurred (42.9%), including 2 cancer deaths and 1 non-cancer death. Cancer progression was not observed during IAC + RT, and the two who died of cancer only showed progression and died after transitioning from IAC + RT to other treatments. This treatment group revealed one death from liver failure. The Kaplan–Meier method was used to analyze the prognosis of BTC treated with IAC + RT. The mOS was 18.1 months, and the mPFS from the start of IAC + RT until disease progression was 10.7 months (Figure 2). The 1-year survival rate was 28.6% (2 cases), but 4 patients remained alive at the end of this study.

### 3.4. Adverse Events

All adverse events of this study are summarized in Table 4. Only grade 1 or 2 levels of renal dysfunction, hypoalbuminemia, nausea, fatigue, and rash occurred, which were not persistent problems. The rate of cytopenia was higher among grade 3 and 4 levels of adverse events than that in previous reports: leukopenia (1170–1740/mm^3^) in 5 (71.4%) cases, neutropenia (610–940/mm^3^) in 5 (71.4%) cases, hemoglobin depletion (7.6–7.6 g/dL) in 2 (28.6%) cases, and thrombocytopenia (35,000–45,000/mm^3^) in 4 (57.1%) cases. There was one case (14.3%) that showed liver dysfunction (7.7 mg/dL of total bilirubin). A grade ≥3 increase in pancreatic enzyme levels (353 U/L of amylase and 102 U/L of lipase) with no symptoms was found but immediately improved in 2 (28.6%) cases. Regarding non-hematological toxicity, cholangitis was most frequent, being noted in 2 (28.6%) cases. Anorexia, abdominal pain, gastroduodenal ulcer, liver abscess, and catheter deviation were observed in one case each. Unexpected serious adverse events or deaths were not observed during the study.

## 4. Discussion

This is the first prospective study to test the antitumor effects and adverse events of IAC + RT as the first-line therapy and suggest that IAC + RT may be an effective treatment option for unresectable BTC with locally advanced liver and lymph node metastases. Patients treated with IAC + RT had an extremely high clinical RR of 71.4%, achieving an RR nearly 30% better than that of previously reported patients treated with systemic chemotherapy.

The high RR of IAC + RT is because of the theoretical usefulness of IAC in its pharmacokinetics. Local anticancer drug injection via IAC was considered as it can administer higher doses of drugs due to the fast pass effect and increase the concentration at the lesion [20]. IAC has been shown to have a high RR in previous reports [21]. The results of this study were consistent with those of previous findings because radiation is considered to have a high RR [16,17]. IAC, which targets not only the main lesion but also liver infiltration and liver metastasis, exerts an antitumor effect in a region different from RT, which targets the main lesion and lymph node metastasis. Therefore, the combination of IAC and RT is expected to achieve a stronger effect than either monotherapy. IAC + RT obtained a high RR not only in a retrospective study but also in a prospective study of different institutions.

We made several adjustments to facilitate the performance of prospective studies based on our retrospective data [19]. This IAC + RT therapy was indicated for patients who had not only a high RR but also a good prognosis in our retrospective study. Specifically, cases with locally advanced or lymph node and liver metastasis were indicated, and cases with distant metastasis or peritoneal dissemination, which were out of the therapeutic range and unlikely to achieve an improved prognosis, were excluded. Off-label drugs, which were previously used as one-shot IAC in our retrospective study, were excluded from the current study. The doses of 5FU and CDDP in r-IAC were unified to 750 mg/m^2^ and 10 mg/m^2^, respectively. A reduction in the RR raises some concern due to the exclusion of certain drugs used in one-shot IAC, but the RR remained high. The frequency of neutropenia and thrombocytopenia increased markedly from 1.9% and 11.5% to 71.4% and 57.1%, respectively, when we compared the grade ≥3 adverse events in this study with those in the retrospective study, but the rate of cholangitis and gastroduodenal ulcer decreased from 23.1% and 25.0%, respectively, to 14.3%. The increased hematological adverse events in comparison to those in the retrospective study have unclear causes. One of the reasons is the small sample size. Otherwise, drug doses were individually adjusted in cases with high risks, such as high age and poor performance status, which may have caused decreased hematological adverse events in our retrospective study. The increased number of hematological adverse events could be safely managed with no protocol-related deaths, and the protocol completion rate was quite high. IAC + RT therapy seems to be acceptable for treating unresectable BTC, with locally advanced liver and lymph node metastases, under sufficient attention for cholangitis and gastroduodenal ulceration as well as cytopenia.

The IAC + RT therapy was compared to systemic chemotherapy, which is the standard treatment of BTC. The RR of systemic chemotherapy is reportedly to be approximately 20–30% [9,10,11,12,22]. Thus, the rate of 57.1–71.4% with this treatment was extremely high in comparison to the rates described in previous reports. The precise evaluation of the OS and PFS was difficult because two cases were unable to continue r-IAC due to the completion of this study in response to the law’s revision. These data are included for reference only, and the mOS and mPFS with this treatment were 18.1 and 10.7 months, respectively, which is considered good compared to the roughly 12 and 6 months, respectively, with GC therapy [9,10]. More than half of the patients remain alive at the end of the study. Thus, further improved prognosis would be highly possible despite additional analyzes. The same trend as in our retrospective study was obtained (mOS: 15.4 months and mPFS: 14.3 months) [19]. Thus, this treatment was considered to have a very high tumor control ability. However, the benefits of this treatment are limited to cases of local progression and lymph node and liver metastases. The treatment effect was not shown in cases of distant metastases in our retrospective and present studies [19].

Immune checkpoint inhibitors have been investigated, such as anti-programmed death 1 (PD-1) or anti-cytotoxic T-lymphocyte antigen-4 (CTLA-4) antibodies [23,24], as other treatments against distant metastasis, but with no significant results confirmed at present. Hence, the combination of durvalumab, which is a PD-L1 (programmed death-ligand 1) inhibitor, and GC standard therapy reduced the risk of death by 20% compared to GC alone (hazard ratio: 0.80, *p* = 0.021), and RR and PFS were also improved in the combination group in the TOPAZ-1 trial [25,26]. The combination of GC and durvalumab may become the new standard first-line treatment for advanced BTC. The MOSCATO-01 trial [27] was the first to show consistent molecular changes with appropriate targeted molecular therapy in a large evaluation of difficult-to-treat cancers [28], including 43 BTCs, suggesting target-rich tumors in BTCs with a high proportion of patients with actionable changes. The novel above-mentioned treatments are expected to be an effective option for the treatment in cases of distant metastases.

The BT-22 study observed the following grade 3 and 4 hematological adverse events: leukopenia in 29.3%, anemia in 36.6%, thrombocytopenia in 39%, and neutropenia in 56.1% [10]. The rate of hematological adverse events in this study (leukopenia in 71.4%, thrombocytopenia in 57.1%, and neutropenia in 71.4%) was even higher than that in BT-22 tests, as determined in a previous study with systemic chemotherapy [10]. This IAC + RT therapy has little effect on the whole body because it was administered at a high concentration, mainly to the primary tumor and its surroundings, but it remained important to care for the adverse effects of the whole body in IAC + RT treatment as well as for those associated with systematic chemotherapy. Cholangitis is caused by biliary obstruction due to biliary sludge or tumor progression. One patient discontinued treatment due to cholangitis, although the antitumor effect was sufficient in this study. Cholangitis due to biliary obstruction can make treatment continuation difficult, even if the tumor is controlled, which is one of the problems with BTC [29]. Cholangitis frequently occurred in IAC + RT (28.6%) as well as that described in the BT-22 study [30]; thus, cholangitis prevention by appropriate biliary drainage is very important in antitumor therapy. Patients with liver dysfunction (7.7 mg/dL of total bilirubin) in this study also experienced difficulty controlling cholangitis, which required multiple stentings by endoscopic drainage. A strict value of jaundice in the treatment criteria is considered necessary for avoiding treatment-associated severe cholangitis. Regarding gastroduodenal ulcers, few reports were caused by systemic chemotherapy, and only one case of grade 3 gastroduodenal ulcer occurred in this study. The true risk of the event should be clarified due to the IAC + RT. IAC + RT revealed high antitumor efficacy when taking care of severe adverse events such as hepatic failure by comparing the data on systemic chemotherapy. The IAC dose should be reduced, and the inclusion criteria should be tightened in the future.

Several limitations associated with the present study warrant mention, including its small sample size due to study termination following the law’s revision. This small number of cases is not sufficient to determine the efficacy and safety of this treatment. Evaluating the second-line treatments and performing accurate mOS analyses and assessments of the hematological adverse events that were more frequent than in the retrospective study has been difficult. We will verify and report the effects of IAC + RT on the prognosis by performing an additional analysis at the end of this research period. We would also like to plan future prospective research with more cases for the additional prognosis and adverse event evaluation as well as second-line treatments, and further large multi-center studies are required to establish an appropriate protocol for IAC + RT.

## 5. Conclusions

In summary, this study revealed for the first time that IAC + RT has a high RR and disease control rate, even in a prospective setting. IAC + RT therapy with its high RR was suggested as potentially useful not only for controlling unresectable BTC but also as a preoperative treatment of locally advanced BTC.

## Figures and Tables

**Figure 1 cancers-15-02616-f001:**
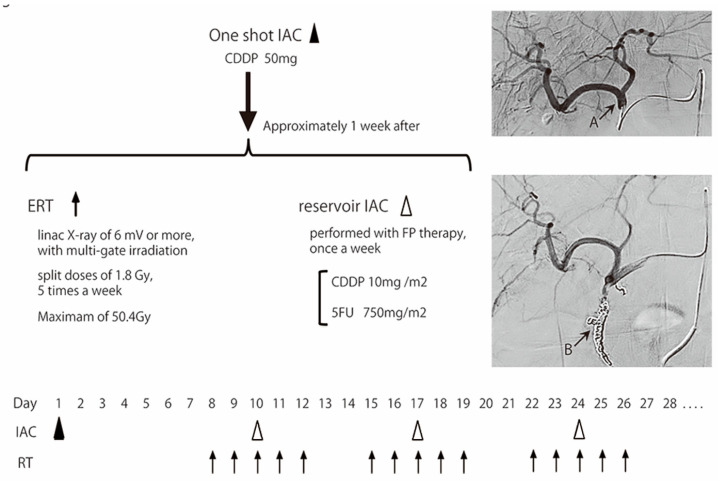
Process of IAC + RT combination therapy.

**Figure 2 cancers-15-02616-f002:**
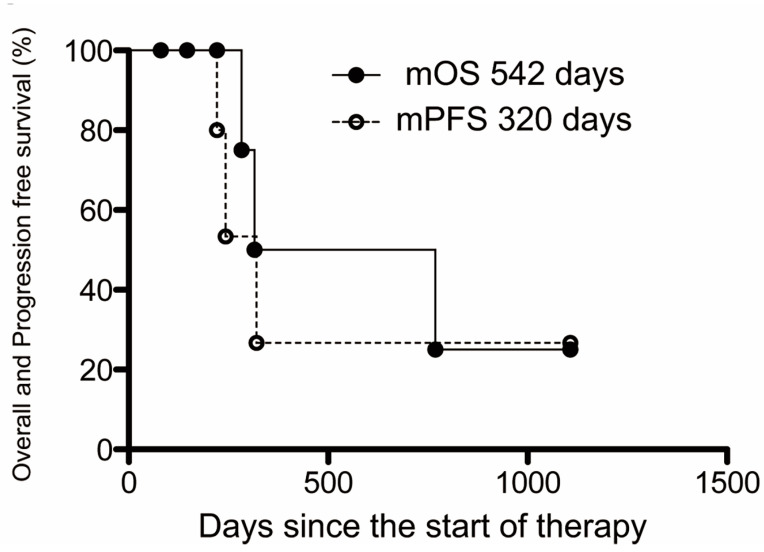
The median overall and progression-free survival of IAC + RT. Kaplan–Meier estimates of the mOS and mPFS. The mOS and mPFS were 542 and 320 days (18.1 and 10.7 months), respectively.

**Table 1 cancers-15-02616-t001:** Baseline characteristics of patients and lesions.

Total, N	Biliary Tract Cancer N = 7
**Age, years**	
Median	71
Range	55–76
**Sex, *n* (%)**	
Female	4 (57.1)
Male	3 (42.9)
**Performance status, *n* (%)**	
0	5 (71.4)
1	2 (28.6)
**Primary site, *n* (%)**	
Gallbladder	3 (42.9)
Perihilar extrahepatic bile duct	2 (28.6)
Distal extrahepatic bile duct	2 (28.6)
**Type of tumor, *n* (%)**	
Adenocarcinoma *	7 (100)
**Stage (UICC 8), *n* (%)**	
Stage 2b	1 (14.3)
Stage 3c	1 (14.3)
Stage 4	H 1 (14.3)
Stage 4b	H 2, N 2 (57.1)
**Jaundice (at diagnosis), *n* (%)**	
No	2 (28.6)
Yes	5 (71.4)
**Jaundice (at the start of treatment), *n* (%)**	
No	7 (100)
Yes	0 (0)
**Child–Pugh scores**	
A	6 (85.7)
B	1 (14.3)
**CEA value, ng/mL (Normal; <5)**	
Median	2.2
Range	1.1–8.7
**CA19-9 value, U/mL (Normal; <37)**	
Median	51
Range	13–4584
**Primary tumor diameter, mm**	
Median	34.2
Range	29.8–52.8
**Total tumor diameter, mm**	
Median	62.2
Range	31.0–96.3

H: liver metastasis; N: extra-area lymph node metastasis. * One case showed partial squamous differentiation.

**Table 2 cancers-15-02616-t002:** Summary of the overall response.

Total, N	Biliary Tract Cancer N = 7
**CR:PR:SD:PD**	RECIST 0:4:3:0, Clinically 1:4:2:0
**Response rate**	RECIST 57.1% (4/7), Clinically 71.4% (5/7)
**Disease control rate**	100% (7/7)

CR: complete response; PR: partial response; SD: stable disease; PD: progression disease; RECIST: Response Evaluation Criteria in Solid Tumors.

**Table 3 cancers-15-02616-t003:** Reduction rate and survival time of each case.

Case	Pre-Treatment Diameter (mm)	Post-Treatment Diameter (mm)	Reduction Rate (%)	Overall Survival (Day)	Progression-Free Survival (Day)
Case 1	78.4	47.4	39.5	283	172
Case 2	45.3	34.2 > 0	24.5 > 100	1107	179
Case 3	96.3	44.7	53.6	769	202
Case 4	31.0	26.1	15.8	315	201
Case 5	29.3	19.8	32.4	221	207
Case 6	69.4	43.3	37.6	146	146
Case 7	62.2	53.2	14.5	80	80

Case 2 includes both RECIST-based and clinical course results with confirmed histological complete response postoperatively.

**Table 4 cancers-15-02616-t004:** Summary of the adverse events.

Total, *n*	Biliary Tract Cancer N = 7	
	All Grades (%)	Grade 3, 4 (%)
**Hematologic**		
Leukopenia	6 (85.7)	5 (71.4)
Neutropenia	6 (85.7)	5 (71.4)
Anemia	6 (85.7)	2 (28.6)
Thrombocytopenia	7 (100)	4 (57.1)
Liver dysfunction	2 (28.6)	1 (14.3)
Pancreatic enzyme elevation	2 (28.6)	2 (28.6)
Renal dysfunction	1 (14.3)	0 (0.0)
Hypoalbuminemia	2 (28.6)	0 (0.0)
**Non-** **hematologic**		
Anorexia	3 (42.9)	1 (14.3)
Abdominal pain	5 (71.4)	1 (14.3)
Nausea	4 (57.1)	0 (0.0)
Diarrhea	0 (0.0)	0 (0.0)
Gastroduodenal ulcer	3 (42.9)	1 (14.3)
Cholangitis	2 (28.6)	2 (28.6)
Fatigue	2 (28.6)	0 (0.0)
Rash	1 (14.3)	0 (0.0)
Liver abscess	1 (14.3)	1 (14.3)
Catheter deviation	1 (14.3)	1 (14.3)

## Data Availability

The datasets used and analyzed during the current study are available from the corresponding author upon reasonable request.

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
