# Peer review of "The Effectiveness of the Combination of Arterial Infusion Chemotherapy and Radiotherapy for Biliary Tract Cancer: A Prospective Pilot Study"

_cancers, 2023, doi:10.3390/cancers15092616_

Round 1

Reviewer 1 Report

Advanced or metastatic biliary tract cancer has exhibited poor response rates to existing therapies, a novel intra-arterial chemotherapy and radiation therapy combination therapy has been used in this study. Two patients groups were allowed to undergo surgery, as responder and disease control rates in a prospective research were 71.4% and 100%, respectively. It could offers promise as a preoperative therapy method for long-term prognosis and novel treatments.

Comments-

These are some experiments that could enhance the quality of the paper:

1-Increase the sample size: Originally conceived as a prospective study including 20 patients, the project was stopped in March 2019 owing to a change in Japanese legislation regarding clinical research. Raising the sample size might increase statistical power and bolster the study's findings.

2-Control group: The article lacked a control group, which may have helped assess the effectiveness of the IAC+RT therapy in comparison to alternative therapies or no treatment. Including a control group might assist demonstrate the therapy's efficacy.

3-Extended follow-up: The CT antitumor effectiveness was only evaluated at three and six months in this study. A extended follow-up period might give more information into the therapy's long-term efficacy.

4-The assessment of the anticancer effectiveness was not conducted blindly. Blinding the evaluator might eliminate any biases and improve the objectivity of the outcomes.

5-The study did not compare IAC+RT to other existing treatments for nonresectable or metastatic breast cancer. Comparing the therapy to different treatment options might aid in identifying the most successful alternative.

6-Comprehensive adverse event reporting: The publication lacked thorough adverse event reporting. Detailing adverse occurrences might aid in identifying possible safety problems and guiding clinical decision-making.

Author Response

Dear Reviewer1:

Thank you very much for your valuable feedback.
Your suggestions have greatly improved our revised manuscript.
Please see our point-by-point responses to your comments below.
Revisions are highlighted in yellow.

Comments-

These are some experiments that could enhance the quality of the paper 

1-Increase the sample size: Originally conceived as a prospective study including 20 patients, the project was stopped in March 2019 owing to a change in Japanese legislation regarding clinical research. Raising the sample size might increase statistical power and bolster the study's findings.

Thank you for your suggestion. We had originally planned to include 20 participants in the study, but increasing the number of study participants was impossible as the study was due to be completed in March 2019, when the law changed. We will make every effort to study an increased sample size based on this report because we need to report data up to March 2019 for the current results.

2-Control group: The article lacked a control group, which may have helped assess the effectiveness of the IAC+RT therapy in comparison to alternative therapies or no treatment. Including a control group might assist demonstrate the therapy's efficacy.

You are correct. We believe that the most accurate data can be presented with a comparative study. However, assembling a sample for comparative study was difficult from the outset as biliary tract cancer is relatively rare. We started with a single-arm study of 20 cases as a realistic number, but unfortunately, the sample size was reduced due to the study’s termination for legal issues. We will make every effort to accumulate more cases based on this report.

3-Extended follow-up: The CT antitumor effectiveness was only evaluated at three and six months in this study. A extended follow-up period might give more information into the therapy's long-term efficacy.

Accordingly, an extended follow-up period is important for long-term efficacy. The mandatory evaluation in the present study is after three and six months, but CT follow-up is performed every 3–6 months until tumor progression. The author would like to present data up to the most recent date if permitted, but the Ethics Committee stipulates data submitted should only be until March 2019. We would be glad to report subsequent data after this report. However, we would appreciate your understanding that we can only present data until March 2019.

4-The assessment of the anticancer effectiveness was not conducted blindly. Blinding the evaluator might eliminate any biases and improve the objectivity of the outcomes.

We believe that a blinded evaluation is more desirable, as you have emphasized. The blind provision was not built at the start of this study. We will keep this in mind for future reference.

5-The study did not compare IAC+RT to other existing treatments for nonresectable or metastatic breast cancer. Comparing the therapy to different treatment options might aid in identifying the most successful alternative.

We have excluded cases of distant metastases from this study because this retrospective study revealed no superiority of IAC + RT therapy in cases of distant metastases. We have added the exclusion criteria in the methods section because understanding the eligible cases is difficult due to insufficient exclusion criteria description. Case groups (local progression, lymph node metastasis, and liver metastasis) for which this treatment is most likely to be recommended were added to the discussion section. Recent advances in recommended treatment in cases of distant metastases have been added to the discussion section for comparison.

6-Comprehensive adverse event reporting: The publication lacked thorough adverse event reporting. Detailing adverse occurrences might aid in identifying possible safety problems and guiding clinical decision-making.

We have added all adverse events, including grades 1 and 2, to Table 4 and the results section, accordingly. We have refrained from presenting numerical values because including all adverse events makes the numerical values difficult to understand. If necessary, please contact us and we will readjust.

Best regard,

Takuma Goto

Reviewer 2 Report

Dr. Goto et al detail a prospective pilot study including 7 BTC patients treated with IAC+RT. My concerns are as follows. 

Major:

1.       A more detailed description is desirable in the PATIENTS section (line 77). Why did the authors include metastatic disease? Does this include distant metastases? Why did the authors include extrahepatic cholangiocarcinoma and ampullary cancer? Please justify these inclusion criteria including references. There is no information on whether previous locoregional treatment or chemotherapy was permitted. Also, were there any exclusion criteria, such as Child-Pugh C or history of recent or co-existing malignancies?
The authors do state in their discussion that some cases of distance metastases were excluded (lines 256-259); this should be in the Methods section.

2.       Was the presence of anatomical variations such as replaced RHA confirmed? If there were any such variations, how was the catheter placed?

3.       I did not understand line 113: “This IAC+RT protocol was defined as ending within six months from its initiation.” Was the protocol terminated after 6 months regardless of the result? Was there a shift to other treatment?

4.       If the last patient had treatment terminated due to changes in the legal environment, shouldn’t that patient be excluded?

5.       There should be more information on the patient who died of liver failure. IAC+RT may have contributed to the liver dysfunction, which can occur months after the end of treatment. Is this the same case that had total bilirubin of 7.7 mg/dl?

6.       I do not believe the authors can conclude that the therapy was safe and tolerable just because there were no treatment-related deaths. Again, I am concerned about point 5 above.

7.       In part because of the early termination, the study is underpowered. The authors cannot state that the treatment is safe or useful based on their results. The authors also should not state that they “demonstrated the benefits over systemic chemotherapy” (line 23).

8.       Table 1: “Bile duct” refers to common bile duct? Please clarify intrahepatic or extrahepatic (perihilar, distal).

9.       Table 1: How was “jaundice” defined? Why weren’t patients with jaundice excluded? Please also provide Child-Pugh scores in this table.

10.   Table 1: Instead of “up to Stage 3” please provide exact stages. Please also provide location of metastases.

11.   Table 1: One cases shoed partial squamous differentiation. This is not adenosquamous carcinoma?

12.   Table 3: Five cases had metastases. Did those grow in size? Were all metastases included within the scope of radiation?

13.   Table 3: Case 2 should not include histological complete response. The authors state they are using RECIST 1.1.

Minor:

1.       Line 19, 30: What do the authors mean by ”main” treatment? Do they mean “standard” treatment?

2.       Line 49: I believe the 5-year survival rate in Japan is currently higher than 10%. The references used are old and recent figures should be confirmed.

3.       Line “according to a retrospective study”: this is the authors’ own work and they should clarify this in the manuscript, to avoid misleading the reader. Also, the reference to the Japanese paper (reference 16) is not necessary if its content is similar to the one in English (reference 17).

4.       Line 84: Instead of “liver-enzyme level” I would write out alanine aminotransferase and aspartate aminotransferase.

5.       Line 177: I would change “leukopenia/neutropenia and cholangitis in 3 cases” to “leukopenia/neutropenia and cholangitis in 3 cases each

6.       Figure 2 is confusing. Please simply provide a table with the necessary numbers. Because there is much overlap with Table 3, the figure can just be deleted.

7.       Abbreviations in the tables should be defined at first use.

Author Response

Dear Reviewer2:

Thank you very much for your valuable feedback.
Your suggestions have greatly improved our revised manuscript.
Please see our point-by-point responses to your comments below.
Revisions are highlighted in yellow.

Major:

1.       A more detailed description is desirable in the PATIENTS section (line 77). Why did the authors include metastatic disease? Does this include distant metastases? Why did the authors include extrahepatic cholangiocarcinoma and ampullary cancer? Please justify these inclusion criteria including references. There is no information on whether previous locoregional treatment or chemotherapy was permitted. Also, were there any exclusion criteria, such as Child-Pugh C or history of recent or co-existing malignancies? The authors do state in their discussion that some cases of distance metastases were excluded (lines 256-259); this should be in the Methods section.

Thank you for your valuable suggestions. We apologize for the lack of clarity due to insufficient description of the exclusion criteria. The metastatic cases indicated are those with liver or lymph node metastases that can be covered by this treatment. Cases with distant metastases were excluded from the study as they did not show better results than standard treatment. Cases with previous chemotherapy are excluded based on the results of the retrospective study. We have added the exclusion criteria to the study protocol. Most of the areas you have emphasized have been included. Please let us know if more corrections are necessary.

2.       Was the presence of anatomical variations such as replaced RHA confirmed? If there were any such variations, how was the catheter placed?

The procedures to be followed in case of anatomical variations, as described in the study protocol, were not described in the initial submission. We have added the content of the protocol to the methods section. Additionally, one case of RHA replacement was observed, and the procedures followed in that case have been added to the results section.

3.       I did not understand line 113: “This IAC+RT protocol was defined as ending within six months from its initiation.” Was the protocol terminated after 6 months regardless of the result? Was there a shift to other treatment?

This treatment is terminated for a maximum of six months, regardless of efficacy, based on the results of a previous report. We have included information on post-treatment after IAC + RT in lines 116–118, but we will correct it if it is unclear.

4.       If the last patient had treatment terminated due to changes in the legal environment, shouldn’t that patient be excluded?

The last case was included in the evaluation because it was also a pilot study, and it completed RT and the IAC lasted almost three months until the end of the study. However, the patient was not treated for a long period, thus we will change the data if the exclusion is appropriate. The patient had SD and the reduction rate was not high, so the absence of this case has not compromised the results.

5.       There should be more information on the patient who died of liver failure. IAC+RT may have contributed to the liver dysfunction, which can occur months after the end of treatment. Is this the same case that had total bilirubin of 7.7 mg/dl?

This is the same case that had total bilirubin of 7.7 mg/dl. Multiple endoscopic biliary drainages were performed due to several bile duct infections caused by hilar bile duct stenosis. The liver failure was thought to be caused by the inability to control the cholangitis because it was not during an IAC. However, the impact of this treatment could not be completely ruled out. Therefore, we have added to the discussion section that cholangitis required multiple biliary drainages in the case of liver dysfunction and that the values for jaundice in the treatment criteria may need to be strictly defined.

6.       I do not believe the authors can conclude that the therapy was safe and tolerable just because there were no treatment-related deaths. Again, I am concerned about point 5 above.

We will soften the wording of well tolerated, as we cannot rule out the possibility that the patient has been affected although not during IAC, as we answered in point 5. We have added in the discussion section that we will consider making the jaundice value in the administration criteria more stringent in cases of repeated cholangitis. Further, this small number of cases is not sufficient to establish the safety of this treatment.

7.       In part because of the early termination, the study is underpowered. The authors cannot state that the treatment is safe or useful based on their results. The authors also should not state that they “demonstrated the benefits over systemic chemotherapy” (line 23).

We do not believe that over-systemic chemotherapy is possible in the sample size of this study. We have presented some unresectable BTC cases because it is not over, but it was a difficult expression to understand, thus we will delete the phrase over systemic chemotherapy itself. In the discussion section, we have added that the effectiveness and safety of this treatment are not sufficient to make a decision.

8.       Table 1: “Bile duct” refers to common bile duct? Please clarify intrahepatic or extrahepatic (perihilar, distal).

It is extrahepatic and has been corrected.

9.       Table 1: How was “jaundice” defined? Why weren’t patients with jaundice excluded? Please also provide Child-Pugh scores in this table.

We caused confusion by presenting whether or not jaundice was present at the time of diagnosis. We have provided jaundice values at diagnosis and at the start of treatment in Table 1 to avoid confusion because jaundice cases were absent at the start of treatment. Child–Pugh scores were also presented in Table 1.

10.   Table 1: Instead of “up to Stage 3” please provide exact stages. Please also provide location of metastases.

We have added the exact stage and the metastatic site to Table 1.

11.   Table 1: One cases shoed partial squamous differentiation. This is not adenosquamous carcinoma?

This case was inoperable, and the histological diagnosis by EUSFNA was adenocarcinoma. The cytology revealed squamous differentiation, but making a diagnosis of adenosquamous carcinoma was difficult. Adenosquamous carcinoma diagnosis may have been possible if the case was operable.

12.   Table 3: Five cases had metastases. Did those grow in size? Were all metastases included within the scope of radiation?

Metastases included the liver and extra-area lymph nodes, which were within the range covered by IAC and the range of radiation, respectively, thus both did not grow in size.

13.   Table 3: Case 2 should not include histological complete response. The authors state they are using RECIST 1.1.

We have presented the RECIST judgement. Clinically obtained effects were described separately.

Minor:

1.       Line 19, 30: What do the authors mean by ”main” treatment? Do they mean “standard” treatment?

We have changed all of them to “standard treatment” as you have indicated.

2.       Line 49: I believe the 5-year survival rate in Japan is currently higher than 10%. The references used are old and recent figures should be confirmed.

We have added the recent information in the Introduction section and changed the reference to a new one.

3.       Line “according to a retrospective study”: this is the authors’ own work and they should clarify this in the manuscript, to avoid misleading the reader. Also, the reference to the Japanese paper (reference 16) is not necessary if its content is similar to the one in English (reference 17).

We have changed “the retrospective study” to “our study” as you indicated. We have also deleted reference 16.

4.       Line 84: Instead of “liver-enzyme level” I would write out alanine aminotransferase and aspartate aminotransferase.

We have made the corrections accordingly.

5.       Line 177: I would change “leukopenia/neutropenia and cholangitis in 3 cases” to “leukopenia/neutropenia and cholangitis in 3 cases each”

We have made the corrections as indicated.

6.       Figure 2 is confusing. Please simply provide a table with the necessary numbers. Because there is much overlap with Table 3, the figure can just be deleted.

We have removed Figure 2 as recommended because its contents overlap with Table 3.

7.       Abbreviations in the tables should be defined at first use.

We have added abbreviations to your remarks.

Best regard,

Takuma Goto

Reviewer 3 Report

In this manuscript, the authors examined the effectiveness and safety of intra-arterial chemotherapy plus radiation therapy as the first-line therapy and found it to be highly effective in cases of unresectable biliary tract cancer. The combination therapy demonstrated the benefits over systemic chemotherapy, leading to a high response rate and disease control rates of 71.4% and 100%, respectively. Moreover, two patients have converted to surgery, indicating a high potential as a preoperative therapeutic strategy to achieve a long-term prognosis. Although the results are potentially interesting, there are one significant issue to be addressed before considering its acceptance.

Major comments:

1. The authors mentioned the Grade ≥3 TRAEs in this study with those in the retrospective study, the frequency of neutropenia and thrombocytopenia increased markedly from 1.9% and 11.5% to 71.4% and 57.1%, respectively. The authors assumed that this change was possibly due to a small number of cases in the pilot study. There were no treatment-related deaths, while the grade ≥3 hematologic TRAEs mostly seems to be increased significantly. Could the authors supplement more explanation to elucidate what’s the reason and how to solve this issue clinically?

Author Response

Dear Reviewer3:

Thank you very much for your valuable feedback.
Your suggestions have greatly improved our revised manuscript.
Please see our point-by-point responses to your comments below.
Revisions are highlighted in yellow.

Major
The authors mentioned the Grade ≥3 TRAEs in this study with those in the retrospective study, the frequency of neutropenia and thrombocytopenia increased markedly from 1.9% and 11.5% to 71.4% and 57.1%, respectively. The authors assumed that this change was possibly due to a small number of cases in the pilot study. There were no treatment-related deaths, while the grade ≥3 hematologic TRAEs mostly seems to be increased significantly. Could the authors supplement more explanation to elucidate what’s the reason and how to solve this issue clinically?

Thank you for your valuable feedback. The load may have been higher in this prospective study, although it is difficult to identify and is only predictive, because treatment was given at approximate doses in a retrospective study, and treatment was sometimes stopped for safety considerations because there were no stipulations. The solution is to, first, increase the number of cases, but the study results suggest to consider reducing the dose of IAC and tightening the administration criteria. We have added this to the discussion section.

Best regard,

Takuma Goto

Round 2

Reviewer 1 Report

I am happy with the responses and comments included in this revised manuscript.

Reviewer 2 Report

I believe the authors have adequately revised their manuscript.